# An Unsupervised Urban Extent Extraction Method from NPP-VIIRS Nighttime Light Data

**Xiuxiu Chen** [1], **Feng Zhang** [1,2,*], **Zhenhong Du** [1,2] and **Renyi Liu** [1,2]

[1]    School of Earth Sciences, Zhejiang University, Hangzhou 310027, China; cxxtribal@zju.edu.cn (X.C.);
       duzhenhong@zju.edu.cn (Z.D.); liurenyi@zju.edu.cn (R.L.)

[2]    Zhejiang Provincial Key Laboratory of Geographic Information Science, Hangzhou 310028, China

[*]    Correspondence: zfcarnation@zju.edu.cn; Tel.: +86-571-8827-3287

**Abstract:** An accelerating trend of global urbanization accompanying various environmental and urban issues makes frequently urban mapping. Nighttime light data (NTL) has shown great advantages in urban mapping at regional and global scales over long time series because of its appropriate spatial and temporal resolution, free access, and global coverage. However, the existing urban extent extraction methods based on nighttime light data rely on auxiliary data and training samples, which require labor and time for data preparation, leading to the difficulty to extract urban extent at a large scale. This study seeks to develop an unsupervised method to extract urban extent from nighttime light data rapidly and accurately without ancillary data. The clustering algorithm is applied to segment urban areas from the background and multi-scale spatial context constraints are utilized to reduce errors arising from the low brightness areas and increase detail information in urban edge district. Firstly, the urban edge district is detected using spatial context constrained clustering, and the NTL image is divided into urban interior district, urban edge district and non-urban interior district. Secondly, the urban edge pixels are classified by an adaptive direction filtering clustering. Finally, the full urban extent is obtained by merging the urban inner pixels and the urban pixels in urban edge district. The proposed method was validated using the urban extents of 25 Chinese cities, obtained by Landsat8 images and compared with two common methods, the local-optimized threshold method (LOT) and the integrated night light, normalized vegetation index, and surface temperature support vector machine classification method (INNL-SVM). The Kappa coefficient ranged from 0.687 to 0.829 with an average of 0.7686 (1.80% higher than LOT and 4.88% higher than INNL-SVM). The results in this study show that the proposed method is a reliable and efficient method for extracting urban extent with high accuracy and simple operation. These imply the significant potential for urban mapping and urban expansion research at regional and global scales automatically and accurately.

**Keywords:** urban extent extraction; nighttime light data; NPP-VIIRS; a pixels-based unsupervised method; spatial context constraints

---

## 1. Introduction

Urbanization is speeding up the consumption of natural resources, energy, and altering land use and cover, which is closely related to almost all aspects of global change [1,2]. The rapid expansion of urban areas has brought about various environmental and urban issues, such as deforestation, farmland decrease, ecological damage, air pollution, water shortage, climate change, traffic congestion, massive and chaotic urban settlements and mismatched infrastructure in recent years [3,4]. Accurate and consistent information on the distribution and extent of urban areas is fundamental to understanding urbanization dynamics, evaluating the impacts of urbanization on environments, and addressing



the issues mentioned above [5,6]. How to accurately and timely update urban extent has become a compelling topic.

Remote sensing technology provides an effective and alternative approach to detect urban extent for repeated surveys of urban growth [7,8]. Many methods and analysis have carried out in urban mapping and urbanization using remote sensing. Nevertheless, there are still some limitations. Of particular concern is how to obtain urban extent at large scale effectively and timely [9]. Extensive research has shown that the nighttime light (NTL) data, capturing information on human activities, can be an effective, economic, and straightforward data source for extracting urban extent [10–13]. NTL data has advantages in urban extent extraction at large spatial scales or high temporal frequencies [14], because it has appropriate spatial and temporal resolution, free access and global coverage [15]. The most commonly used NTL data is from the Defense Meteorological Satellite Program—Operational Linescan System (DMSP-OLS), which has a relatively long time series released from 1992 to 2013 [16]. A considerable research has grown up around the way to obtain urban extent using DMSP-OLS NTL data. An alternative NTL data provided by the Visible Infrared Imaging Radiometer Suite Day/Night Band carried by the Suomi National Polar-orbiting Partnership (NPP-VIIRS) is available from April 2012 to present [17]. It has a better performance with finer spatial resolution (500 m), lower radiance detection limits, wider radiometric range, and on-board calibration, which significantly reduces the saturation and over-glow problem inherent to DMSP-OLS NTL data [18]. Shi et al. [19] evaluated the potential of NPP-VIIRS NTL data for extracting urban extent and found that the urban extent extracted from NPP-VIIRS NTL data has higher spatial accuracies than that extracted from DMSP-OLS NTL data. In recent years, there has been an increase interest in extracting urban extent using NPP-VIIRS NTL data. However, the research on methods has tended to focus on evaluating the applicability of the methods designed for DMSP-OLS NTL data to NPP-VIIRS NTL data [20], rather than developing methods based on the characteristics of NPP-VIIRS NTL data.

The vast majority of previous research on urban extent extraction methods based on NTL data has focused on threshold techniques and supervised classification techniques. The threshold-based method is most popular because of its simplicity [17]. It selects the appropriate threshold with the auxiliary information, and set pixels greater than the threshold as urban areas and the rest as nonurban areas according the distribution law of the nighttime light radiation [19]. The way of determining the threshold value categorized into three types including empiricism-based [21], mutation-based [11], and reference-based [22,23] methods. The empirical threshold method requires certain expert knowledge or experience to determine the threshold, which is easy to be subjective, resulting in poor accuracy and reliability. The mutation threshold method believes that the real urban areas is composed of complete patches, and the threshold point is the mutation point of patch rupture. The obvious drawback is that it is not suitable for complex urban structures, such as polycentric cities. The reference threshold method utilized statistical data or higher resolution remote sensing images as reference data [19] and takes the pixel value with the minimum urban area difference between the extracted results and the reference data as the threshold for a city. Among them, the threshold method based on high-resolution remote sensing images has better accuracy and reliability, and is widely used in urban extent extraction. It is not appropriate to apply a global threshold for large areas, because thresholds of urban extent extraction varies with socioeconomic development levels [24]. The local optimal threshold (LOT) method has been used in recent studies, which estimates optimal thresholds for each city or tile at a regional scale based on the relationships between nightlight magnitude in NTL data and urban morphology interpreted from high-resolution images [9,18,24,25]. Nevertheless, these relationships and parameters change across regions and vary by time, which requires a large amount of work on interpreting from high resolution images [14]. Unlike the threshold-based method, the supervised classification method regards the process of urban extraction from NTL data as the problem of classifying urban and non-urban pixels [26]. The supervised machine learning has been applied to classify urban and non-urban pixels in NTL data with training set [27]. Cao et al. [26] developed a support vector machine (SVM)-based to extract urban areas from NTL data achieving

comparable results to the LOT method. Xu et al. [28] demonstrated that ANN could provide an effective and accurate alternative in extracting urban built-up areas from NTL. Liu et al. [29] explored the effectiveness of commonly used machine learning methods for urban extent extraction from NTL data, including random forest (RF), gradient boosting machine (GBM), neural network (NN), and their ensemble (ESB), and the results showed that these machine learning methods can achieve similar high accuracies. He et al. [30] proposed a fully convolutional network (FCN) to extract urban land using NTL data and employed it to detect global urban expansion from 1992–2016. The supervised classification method avoids the cumbersome process of threshold selection and raises the automaticity to extract urban extent from NTL data [26]. However, it is difficult to label enough urban pixels. The extracted results using the supervised classification are sensitive to the quality of samples [31].

The previous methods rely highly on the auxiliary data or urban training samples, which largely increases the uncertainty of the results and limits its application in practice, especially for NPP-VIIRS NTL data with higher spatial resolution and radiometric resolution. We attempted to develop an automatic method based on unsupervised methods to improve the efficiency of NTL data in urban mapping. The new method, designed for NPP-VIIRS NTL data, assigned a label to a pixel-based on the light intensity of the pixel and the pixels that are spatially close to it. Clustering is an unsupervised learning method widely used for classification and segmentation of remote sensing images, which was used in this study [32]. Twenty-five cities with different levels of the natural environment and economic development in China were selected as evaluation areas. The urban extent derived from the Landsat8 OLI image was used as the reference for accuracy assessment, and the most commonly used methods the local-optimized threshold method (LOT) and the integrated night light, normalized vegetation index, and surface temperature support vector machine classification method (INNL-SVM) [33] were used to compare, the validity, reliability, and robustness of our method are verified.

## 2. Study Area and Data

### 2.1. Study Area

This study took China as the experimental area because China has experienced rapid urbanization in the last four decades and it has a vast territory with diverse natural conditions and an uneven geographical pattern of socioeconomic development (Figure 1). To evaluate the effectiveness and robustness of the proposed method for urban extent extraction using NPP-VIIRS NTL data, 25 representative cities, located from east to west and from north to south across China, were selected as evaluation areas. The populations of these selected cities varied from 220 thousand to over 21 million and the gross domestic product (GDP) ranged from approximately 20 billion RMB to over 2400 billion RMB (Table 1). These cities, with various urban sizes, diverse physical environments, and different economic and social development levels, can assess the applicability of the proposed urban extent extraction method for different urbanization levels well.

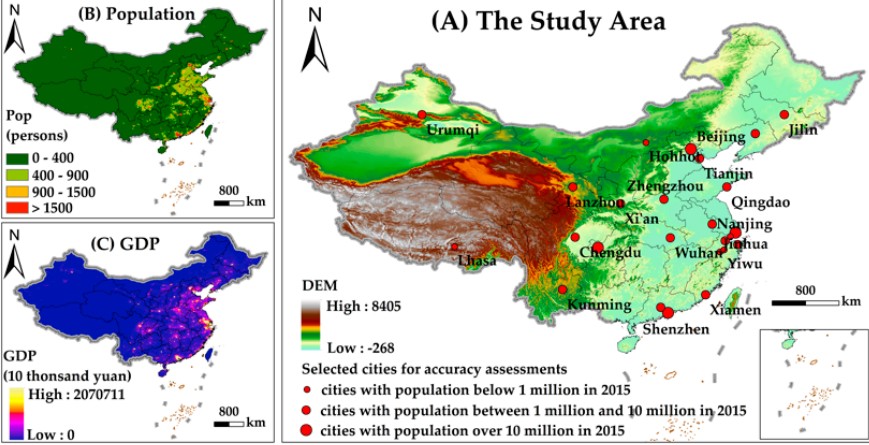

**Figure 1.** *Cont.*

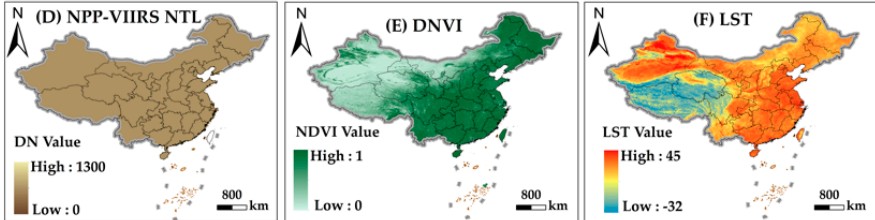

**Figure 1.** Study area: (**A**) The spatial distribution of 25 evaluation areas; (**B**) 1 km grid population data of China in 2015; (**C**) 1km grid gross domestic product (GDP) data of China in 2015; (**D**) The 2015 NPP-VIIRS NTL data of China; (**E**) The 2015 normalized difference vegetation index (NDVI) data of China; (**F**) The nighttime land surface temperature (LST) data of China in 2015.

**Table 1.** The population (Pop) and GDP for Districts under City of the selected cities for 2015 ("Districts under City" includes all the city districts, not including counties and the city at lower levels).

| City | Pop (Million Persons) | GDP (Billion RMB) | City | Pop (Million Persons) | GDP (Billion RMB) |
|---|---|---|---|---|---|
| Beijing | 13.39 | 2301.46 | Shanghai | 13.73 | 2483.84 |
| Hohhot | 1.29 | 230.09 | Nanjing | 6.51 | 972.08 |
| Tianjin | 10.22 | 1653.82 | Hangzhou | 5.16 | 872.20 |
| Qingdao | 3.72 | 597.71 | Ningbo | 2.31 | 487.72 |
| Shenyang | 5.29 | 589.12 | Jiaxing | 0.87 | 87.09 |
| Jilin | 1.82 | 141.38 | Jinhua | 0.96 | 64.57 |
| Xian | 6.04 | 513.64 | Yiwu | 0.77 | 104.51 |
| Urumqi | 2.61 | 261.01 | Wuhan | 5.15 | 880.60 |
| Lanzhou | 2.05 | 174.15 | Zhengzhou | 3.39 | 408.04 |
| Lhasa | 0.22 | 19.69 | Guangzhou | 8.48 | 1810.04 |
| Chongqing | 21.27 | 1320.63 | Shenzhen | 3.44 | 1750.29 |
| Chengdu | 6.93 | 846.00 | Xiamen | 2.07 | 346.60 |
| Kunming | 2.78 | 307.29 | | | |

## 2.2. Data

(1)   Remote Sensing Data

The Earth Observations Group (EOG) is producing a version 1 suite of average radiance composite images using nighttime data from the Visible Infrared Imaging Radiometer Suite (VIIRS) Day/Night Band (DNB). The version 1 series of monthly composites and annual composites is now publicly available through the Payne Institute for Public Policy at the Colorado School of Mines (https://payneinstitute.mines.edu/eog/). The products are 15 arc-second geographic grids, spanning the globe from 75N to 65S in latitude with six tiles. The DN unit is nanoWatts·cm$^{-2}$·sr$^{-1}$. The geographic coordinate system is WGS-1984. Prior to averaging, the NTL data is filtered to exclude data impacted by stray light, lightning, lunar illumination, and cloud-cover. The monthly composites have not been filtered to screen out lights from aurora, fires, boats, and other temporal lights, while the annual composites have layers with additional separation, removing temporal lights and background (non-light) values. We downloaded the 2015 annual NPP-VIIRS NTL composite data and made an outlier removal process to filter out moonlight reflected from water by setting water pixels value to zero.

The Geospatial Data Cloud website (http://www.gscloud.cn/) provides Moderate-resolution Imaging Spectroradiometer (MODIS) daily, 5-days, 10-days, and monthly composite images of normalized difference vegetation index (NDVI) and land surface temperature (LST) covering the whole of China. The monthly composite images of NDVI (MODND1M) of China in 2015 were obtained and processed by maximum value composition (MVC) to create a yearly maximal NDVI image [33]. We also applied the MVC method to the monthly nighttime LST (MODLT1M.LTN) product of China from January to December 2015 to create a yearly maximal LST image.

The 2015 Landsat8 OLI images covering 25 cities with less than 1% cloud cover were obtained from the Geospatial Data Cloud website. Radiation calibration, atmospheric correction, mosaic, clipping, and other pre-processing operations were performed on Landsat8 OLI images of each city with ENVI.

The FLAASH (Fast Line-of-sight Atmospheric Analysis of Spectral Hypercubes) tool was applied to atmospheric correction. The impervious surface was initially classified using maximum likelihood method. Then the actual urban extent was acquired by visually interpreted with the help of 2015 google history images at the fifteenth layer. The spatial resolution of google history images in the 15th layer is 3.66 m, which we downloaded by 91 Weitu (http://www.91weitu.com).

All images were resampled to a spatial resolution of 500m on the Albers Conical Equal Area projection.

(2)    Auxiliary Data

The administrative boundaries of Chinese cities were applied to partition the research areas into cities. The images were cutting out by the extent of each city's administrative boundary. The water mask images excluded the water pixels from the nighttime light images to remove the effect of water on night light. The administrative boundaries and water mask were obtained from the Resource and Environment Data Cloud Platform website (http://resdc.cn/). The 1 km grid population dataset [34] and 1 km grid GDP dataset [35] of China in 2015 were from the Resource and Environment Data Cloud Platform. The statistical data of Table 1 was from the China City Statistical Yearbook 2016. The China City Statistical Yearbook 2016 covered the main socio-economic statistical data of cities at all levels for 2015.

(3)    Other Global Urban Data Products

Recently, there are many global or national urban/impervious surface products released. We chose four existing products for comparison purpose, including the annual maps of global artificial impervious area (GAIA) [36], the high-resolution multi-temporal mapping of global urban land based on a normalized urban areas composite index (NUACI-based GUL) [37], the global urban expansion product based on a fully convolutional network (FCN) [30] and the MCD12Q1.006 MODIS Land Cover Type Yearly Global 500 m product (MODIS500). The MODIS500 were obtained from Google Earth Engine. We extracted the "urban and built-up lands" type for comparison. Because the 2015 urban mapping data is not available from the FCN, we replaced with 2016 FCN data. The spatial resolution of the GAIA and the NUACI-based GUL was 30m. The original spatial resolution was maintained in urban spatial comparison and area calculation. All urban products were resampled to a spatial resolution of 500 m on the Albers Conical Equal Area projection for accuracy assessment.

## 3. Method

The NTL data records the light intensity emitted by objects on the ground in the night, helping to distinguish bright urban areas from dim background [2]. The key to extracting urban extent based on NTL data is to identify urban areas according to the spatial distribution pattern of nighttime light intensity. The clustering method, widely used for classification and segmentation of remote sensing images as an unsupervised approach, makes inferences from input data without referring to known, or labeled, outcomes [38]. This study applied the clustering method based on pixels to extract urban extent from NTL data. The clustering method regarded input pixels as an out-of-order dataset and ignored the spatial context information of pixels. This resulted in the clustering results that were sensitive to noise and outliers. In addition, adjacent pixels in an NTL image were interdependent, and a large part of NTL radiation of a pixel came from the surroundings. Therefore, the spatial context information of each pixel was considered to improve the classification accuracy of urban pixels. There was a roughly negative correlation between nighttime light intensity and distance from the urban center. It was easy to distinguish between urban inner pixels and non-urban inner pixels because of their distinct brightness differences. However, it was difficult to classify urban pixels in urban edge district because of the complex spatial distribution of light intensity and the large light intensity range in urban edge district. Therefore, extracting urban areas separately by urban inner pixels and urban edge pixels was more efficient and suitable.

Therefore, we proposed an automatic method for extracting urban extent from NTL data based on spatial context constraints and clustering algorithm. We first segmented the NTL image into urban interior district, urban edge district and non-urban interior district by a spatial context constrained clustering algorithm. We then classified urban pixels in urban edge district by developing a direction-based spatial context constrained clustering algorithm. Finally, the full urban extent was obtained by merging the urban inner pixels and the urban pixels in urban edge district. In addition, we assessed the performance of our method by evaluating the accuracy of the extracted results and comparing it with two commonly used urban extent extraction methods based on NTL data. The workflow of the proposed method is shown in Figure 2.

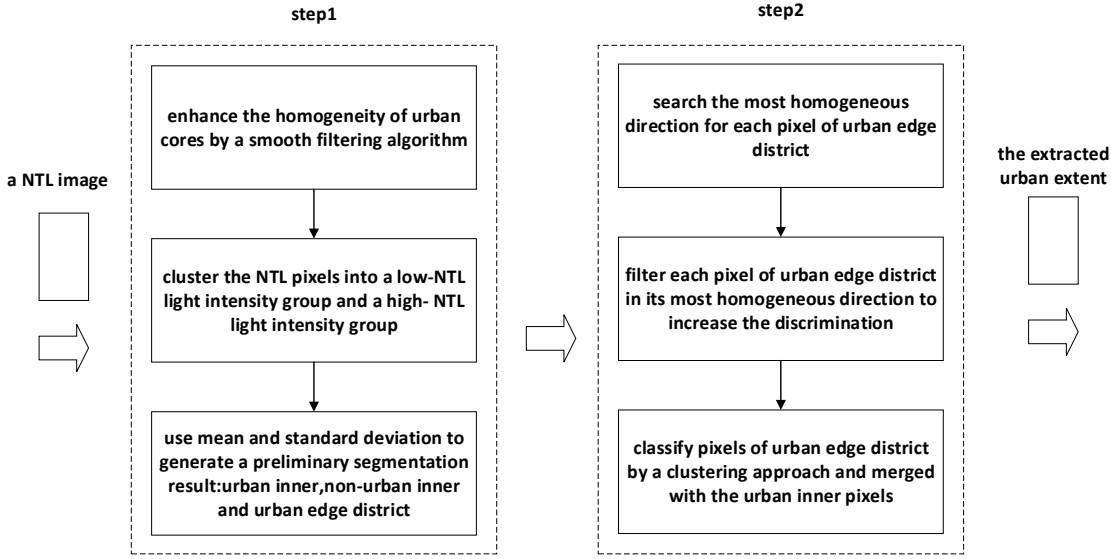

**Figure 2.** The workflow of the proposed method.

### 3.1. The Spatial Context Constrained Clustering Algorithm

K-means clustering is a simple and popular unsupervised machine learning algorithm that partitions data points into k clusters by allocating each point to the cluster with the nearest cluster centroid. The silhouette method plays a significant role in k-means clustering evaluation and help to find the optimum number of clusters [39]. The Silhouette Coefficient is calculated using the mean intra-cluster distance and the mean nearest-cluster distance for each point data in the silhouette method. A higher average Silhouette Coefficient indicates a better clustering effect. Interestingly, when performing K-Means clustering on NTL pixels for different cities, the average Silhouette coefficient always has the highest value when the number of clusters is 2 (Figure 3). Therefore, in this study, we used the K-Means clustering algorithm to classify the NTL pixels into two groups: a low-NTL light intensity group and a high-NTL light intensity group. We assigned pixels in the high- NTL light intensity group are urban pixels and pixels in the low-NTL light intensity group are non-urban pixels.

A sliding window was used to define the surrounding neighborhood. Take the pixels in the window of $(2 \times R + 1) \times (2 \times R + 1)$ with radius R, centered on the pixel $(x, y)$ as the spatial context of the pixel $(x, y)$. We applied the filtering algorithms to integrate the spatial context information to correct, enrich and constrains the current pixel. The adopted filtering algorithms could be mean filtering, median filtering, and other filtering methods, which were determined according to the spatial distribution of light intensity. Then input the filtered pixels into the K-means algorithm for clustering to get the urban pixels. The loss function of the spatial context constrained clustering algorithm was defined as follows.

$$\text{E} = \sum_{j=1}^{K=2} \sum_{p(x,y) \in C_j} \| F(x, y) - \mu_j \|^2, \tag{1}$$

$$\mu_j = \frac{1}{d_j} \sum_{p(x,y) \in C_j} F(x,y), \tag{2}$$

$$F(x,y) = f\left(\left\{N_{(x,y)}\right\}\right). \tag{3}$$

In Equations (1)–(3), E is the loss function; K is the total number of clusters, this study specified K as 2; $C_j$ is the collection of jth cluster; $d_j$ and $\mu_j$ is the number of pixels and the center of the jth cluster, respectively; $f$ is the adopted filtering algorithm and $F(x,y)$ is the filtered result of pixel in $(x,y)$; $\left\{N_{(x,y)}\right\}$ represents the set of all pixels in the sliding window centered on the pixel $(x,y)$.

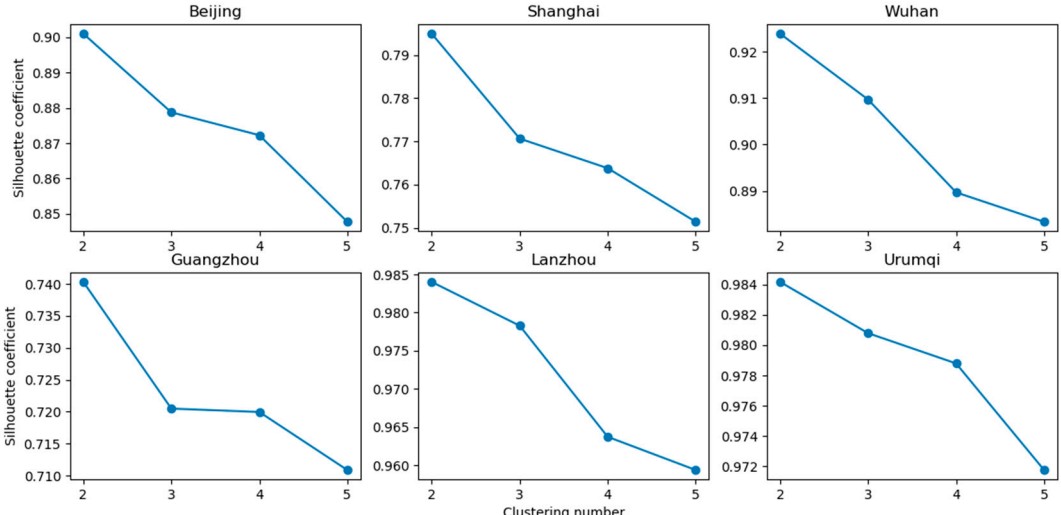

**Figure 3.** Average silhouette coefficient versus number of clusters.

### 3.2. Urban Edge District Detection

We first grouped the NTL pixels into two clusters and regarded the cluster with high light intensity as potential urban ($P_{urb}$) areas and the other as potential non-urban ($P_{non}$) areas. Due to the low light intensity of water, green spaces, and inactive areas at night within urban areas, it was easy to be mistaken as non-urban pixels. We applied the median filter to smooth areas in the urban inner. The median filter is a well-known nonlinear filter with good performance for removing noise, especially for some specific noise types such as "Gaussian", "random", and "salt and pepper" noises. It sorts all the pixel values from the surrounding neighborhood and replaces the current pixel value with the middle value. Compared to the mean filter, the median filter is less affected by the background and better preserves edges while removing noise (Figure 4). The filtered pixels were grouped into $P_{urb}$ areas and $P_{non}$ areas with k-means.

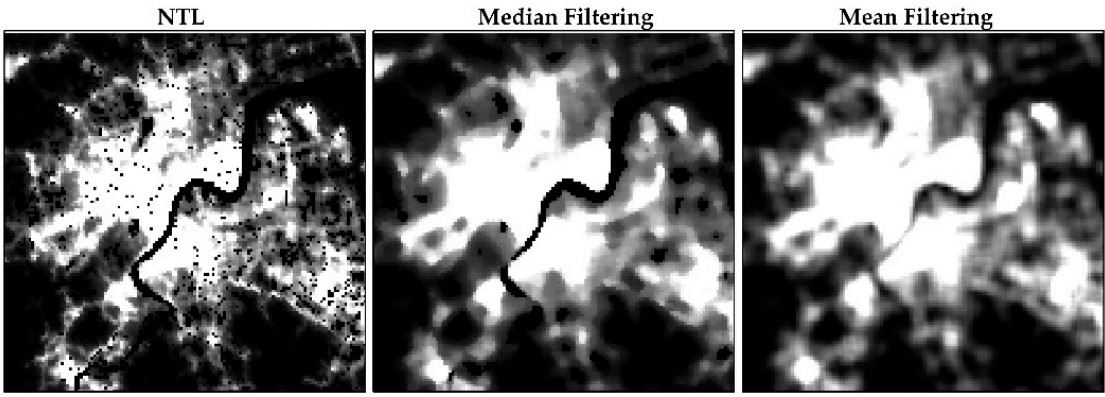

**Figure 4.** Comparison of NTL image filtering results: median filtering and mean filtering.

The spatial distribution pattern of nighttime light intensity corresponds to the urban spatial pattern. In NTL images, the urban core areas are brighter than the surrounding areas, while the countryside and background are dim. Thus, pixels with relatively low light intensity in $P_{urb}$ areas were more likely to be in urban edge district; similarly, pixels with relatively high light intensity in $P_{non}$ areas were more likely to be in urban edge district. The mean value and standard deviation of $P_{urb}$ areas were calculated to define the threshold t2. $P_{urb}$ areas were divided into urban inner areas and edge areas (Equations (4) and (6)). Similarly, $P_{non}$ areas were divided into non-urban inner areas and edge areas by the threshold t1 defined by its mean and standard deviation (Equations (5) and (6)). The urban edge district was obtained by merging the edge areas of $P_{urb}$ areas and $P_{non}$ areas (Equation (6)).

$$P_{urban\ inner} = pixels\ in\ P_{urb}\ , pixel\ value >\ t2 \tag{4}$$

$$P_{non-urban\ inner} = pixels\ in\ P_{non}\ , pixel\ value\ < t1 \tag{5}$$

$$P_{edge\ areas} = \begin{cases} pixels\ in\ P_{urb}\ , pixel\ value \leq t2 \\ pixels\ in\ P_{non}\ , pixel\ value \geq t1 \end{cases} \tag{6}$$

$$t1 =\ P_{non-urb\_mean} + N \times P_{non-urb\_std} \tag{7}$$

$$t2 =\ P_{urb\_mean} - N \times P_{urb\_std} \tag{8}$$

In Equations (4)–(8), $P_{urb\_}mean$ and $P_{urb\_}std$ is the mean value and the standard deviation of $P_{urb}$ areas, respectively; $P_{non-urb\_}mean$ and $P_{non-urb\_}std$ is the mean value and the standard deviation of $P_{non}$ areas, respectively; $P_{urban\ inner}$ and $P_{non-urban\ inner}$ is the pixels within the urban inner areas and the non-urban inner areas, respectively; $P_{edge\ areas}$ is the pixels of the urban edge district; t1 and t2 are the segment thresholds for $P_{non}$ areas and $P_{urb}$ areas, respectively, which are determined by the mean value, the standard deviation of the clusters. N is a coefficient, usually 1.

### 3.3. Urban Pixels Recognition in the Urban Edge District

The urban and non-urban pixels at the edge district have similar and confusing nighttime light intensity, and the urban edge pixels contain more complex meanings and neighborhood spatial heterogeneity than the non-edge pixels. It cannot simply treat the urban edge pixels as noise or one class and requires mining deep spatial information in a larger neighborhood space. Besides, the spatial distribution of light intensity in the edge district presents a certain directionality. We first used an adaptive directional filter algorithm to integrate a larger spatial neighborhood information and filter interference information at the same time, which could enhance the discrimination of urban and non-urban pixels. We then clustered the filtered result into two groups with K-Means. Finally, pixels in a high-intensity group were classified as urban pixels.

Direction filter replaced the center pixel value with the average value of the pixels in the considered direction within the window. We considered eight directions including east, south, west, north, southeast, northeast, southwest, and northwest. The eight direction templates were showed as Figure 5. The filtering direction discrimination function was as Equations (9) and (10). $S_i$ was the standard deviation of all available pixel values in the i-th direction template centered on the current pixel. The filtering direction was satisfied Equations (9), which was selected the direction template with the minimum standard deviation.

$$Dir = min\{S_1,\ S_2, S_3, \ldots, S_8\}, \tag{9}$$

$$S_i = \sqrt{\frac{\sum_{j=1}^{r+1}\left(p_j -\ \bar{p}_i\right)^2}{r}}, \tag{10}$$

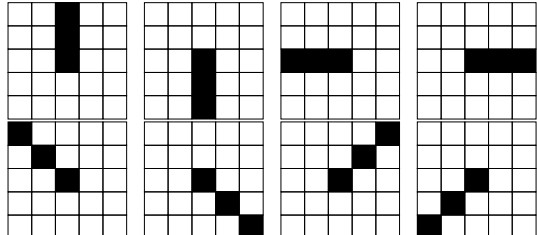

**Figure 5.** Eight filtering direction templates.

### 3.4. Accuracy Evaluation

According to the previous studies, the most popular way of conducting the accuracy evaluation is to compare the extracted urban extent with those derived from Landsat 8 OLI images [19,26,33]. Therefore, we used urban extent obtained from Landsat8 OLI images as the reference for accuracy evaluation. We defined urban areas as human settlements and functionally-related regions that have high density of roads, buildings, shops, and restaurants that attract people, including suburban regions closely related to urban cores. Firstly, we classified the impervious surface from Landsat8 OLI images and produced discrete and bounded patches having a high percentage of impervious surface. Then the coarse urban extent was derived through recognizing and adjusting those patches under visually interpreting Landsat8 OLI images. For those areas that were not easy to recognize, we interpreted with the help of google high-resolution images and POIs to interpret, and got the finer urban extent that was regarded as the actual urban extent.

The confusion matrix was calculated by comparing the urban extent extracted from the NTL data with the reference data pixel by pixel. Quantified the performance of the urban extraction methods with four accuracy indexes, the overall accuracy (OA), the kappa coefficient, the omission error (OE) and the commission error (CE). The specific calculation formula of evaluation index is as follows.

OA is the proportion of correctly classified pixels to the total number of pixels, including correctly classified urban pixels and correctly classified non-urban pixels.

$$OA = \frac{Urban_{true} + Non\_Urban_{true}}{N}, \tag{11}$$

Kappa coefficient indicates the consistency between the extracted results and the reference data, and statistically reflects the extent to which the classification results are better than the random classification results. Pe is the proportion of misinterpretations caused by accidental factors.

$$\text{Ka} = \frac{OA - Pe}{1 - Pe}, \tag{12}$$

$$\text{Pe} = \frac{Urban_{pre} \times Urban_{real} + Non\_Urban_{pre} \times Non\_Urban_{real}}{N \times N}, \tag{13}$$

OE is the proportion of prediction errors in city pixels, reflecting the underestimation of extracted results. The producer's accuracy (PA) is the recognition rate of the actual urban pixels. The sum of OE and PA is 1.

$$OE = \frac{Non\_Urban_{false}}{Urban_{true} + Non\_Urban_{false}}, \tag{14}$$

CE is the proportion of prediction errors in the pixels predicted as cities, reflecting the overestimation of the extracted results. The user's accuracy (UA) is the correct proportion of the predicted urban pixels. The sum of CE and UA is 1.

$$CE = \frac{Urban_{false}}{Urban_{true} + Urban_{false}}, \tag{15}$$

N is the total number of the input pixels. *Urban_real* and *Non_Urban_real* are the number of urban pixels and the number of non-urban pixels in reference data, respectively. *Urban_pre* is the number of urban pixels in extracted result, and *Non_Urban_pre* is the number of non-urban pixels in extracted result. *Urban_true* is the number of pixels that are the urban pixels in the extracted result and are also the urban pixels in the reference data. *Urban_false* is the number of urban pixels in the extracted result which are non-urban pixels in the reference data. *Non_Urban_false* is the number of non-urban pixels in extracted result which are urban pixels in the reference data.

## 4. Results

### 4.1. Selection of Neighborhood Size

We tested various neighborhood size for urban edge district detection and urban edge pixels classification (Figure 6). We found that the larger the neighborhood size for urban edge district detection, the lower PA and the higher UA of the extracted results were, and the Kappa coefficient reached the highest value in a certain range. It was similar with the neighborhood size for urban edge pixels classification. By comparing the filtering and clustering effects with different neighborhood sizes, we used $5 \times 5$ window size to detect urban edge district and $9 \times 9$ window size to classify urban edge pixels.

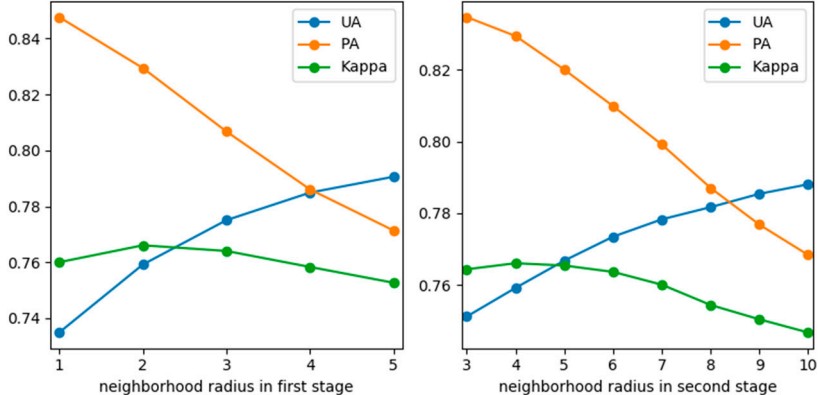

**Figure 6.** Extraction accuracy of urban extent under different neighborhood sizes: on the left is the different neighborhood sizes of the first stage (fixed neighborhood radius 4 in the second stage); on the right is the different neighborhood sizes of the second stage (fixed neighborhood radius 2 in the second stage).

### 4.2. Accuracy and Comparison

The urban extent for 25 cities were extracted and compared using the proposed method, the LOT method, and the INNL-SVM method. The urban extent extraction process of the LOT method and the INNL-SVM method were referred to Dou et al. [33]. Then we calculated the confusion matrix of these three methods between the extracted results and the actual urban extent extracted from Landsat8 OLI, respectively. OA, kappa coefficient, OE, and CE of three methods were as shown in Table 2. The average OA of our method, LOT and INNL-SVM, were 0.9625, 0.9622, and 0.9571, and the average kappa coefficients were 0.7679, 0.7544, and 0.7304, respectively. The method had the highest average kappa coefficient and the second highest average of OA very close to LOT. As a whole, the three methods had high classification accuracy. Since there was a large number of non-urban background pixels in the input data, which can greatly improve the value of OA, we mainly compared the kappa coefficients of the three methods. The lowest kappa coefficients of our method, LOT and INNL-SVM were 0.6871, 0.6645, and 0.6083, respectively. Kappa coefficient over 0.61 means that the results is highly consistent with the reference data. Therefore, the extracted results of our method had high reliability. The kappa coefficients of the extracted results by our method were mostly higher than

LOT and INNL-SVM in 25 cities, which indicated that our method had strong robustness and better performance. We used OE and CE to evaluate the underestimation and overestimation of the three methods, respectively. The higher value of OE, the more serious of underestimation, and the higher value of CE, the more serious of overestimation. For most cities, the OE and CE of the three methods were between 0.1 and 0.35. The results of LOT had similar OE and CE in most cities. The OE and CE of INNL-SVM varied greatly in different places. It meant that the performance of INNL-SVM was greatly affected by different regionally environment. The OE of our method was less than CE in most cities, except in Hohhot, Lanzhou, Lhasa, and Shenyang. Our method had OE greater than CE in these four cities. This was because human activities in these four cities were mainly concentrated urban interior areas, while urban fringe areas were more prone to underestimation. In most cities, the OE of our method was smaller than that of LOT and INNL-SVM, while CE was generally between LOT and INNL-SVM.

**Table 2.** Accuracy assessments of the three methods of extracting urban extent by city.

| City | Our Method | | | | LOT | | | | INNL-SVM | | | |
|---|---|---|---|---|---|---|---|---|---|---|---|---|
| | OA | Kappa | OE | CE | OA | Kappa | OE | CE | OA | Kappa | OE | CE |
| AVG | 0.9625 | 0.7679 | 0.1714 | 0.2373 | 0.9622 | 0.7544 | 0.2209 | 0.2254 | 0.9571 | 0.7304 | 0.2255 | 0.2413 |
| STD | 0.0264 | 0.0384 | 0.0720 | 0.0700 | 0.0261 | 0.0403 | 0.0451 | 0.0443 | 0.0308 | 0.0577 | 0.1141 | 0.0969 |
| Beijing | 0.9563 | 0.7978 | 0.1397 | 0.2118 | 0.9522 | 0.7704 | 0.2020 | 0.2030 | 0.9523 | 0.7539 | 0.2808 | 0.1469 |
| Hohhot | 0.9915 | 0.7573 | 0.3409 | 0.0985 | 0.9914 | 0.7858 | 0.2117 | 0.2078 | 0.9917 | 0.7794 | 0.2717 | 0.1520 |
| Jinhua | 0.9736 | 0.7065 | 0.2098 | 0.3384 | 0.9727 | 0.6687 | 0.3170 | 0.3170 | 0.9685 | 0.6765 | 0.1747 | 0.4032 |
| Shenyang | 0.9741 | 0.7589 | 0.2423 | 0.2119 | 0.9740 | 0.7620 | 0.2242 | 0.2242 | 0.9538 | 0.6782 | 0.0640 | 0.4389 |
| Jilin | 0.9963 | 0.7573 | 0.2272 | 0.2541 | 0.9966 | 0.7687 | 0.2296 | 0.2296 | 0.9968 | 0.7860 | 0.2067 | 0.2180 |
| Xian | 0.9734 | 0.7961 | 0.1385 | 0.2350 | 0.9733 | 0.7837 | 0.2016 | 0.2025 | 0.9765 | 0.8022 | 0.2199 | 0.1476 |
| Urumqi | 0.9905 | 0.8263 | 0.1606 | 0.1768 | 0.9898 | 0.8113 | 0.1835 | 0.1835 | 0.9894 | 0.8211 | 0.0908 | 0.2423 |
| Lanzhou | 0.9945 | 0.8256 | 0.2029 | 0.1378 | 0.9925 | 0.7848 | 0.1590 | 0.2577 | 0.9906 | 0.7493 | 0.1342 | 0.3322 |
| Tianjin | 0.9496 | 0.7749 | 0.1715 | 0.2195 | 0.9450 | 0.7476 | 0.2211 | 0.2209 | 0.9111 | 0.6613 | 0.1190 | 0.4032 |
| Qingdao | 0.9685 | 0.7236 | 0.1669 | 0.3341 | 0.9683 | 0.6893 | 0.2940 | 0.2940 | 0.9660 | 0.6419 | 0.3890 | 0.2834 |
| Shanghai | 0.9165 | 0.7879 | 0.0840 | 0.2167 | 0.9125 | 0.7650 | 0.1769 | 0.1769 | 0.9034 | 0.7533 | 0.1182 | 0.2360 |
| Nanjing | 0.9460 | 0.7801 | 0.1360 | 0.2350 | 0.9458 | 0.7674 | 0.2013 | 0.2013 | 0.9393 | 0.7514 | 0.1681 | 0.2540 |
| Hangzhou | 0.9677 | 0.7443 | 0.1657 | 0.2996 | 0.9674 | 0.7194 | 0.2627 | 0.2637 | 0.9661 | 0.6707 | 0.3942 | 0.2031 |
| Ningbo | 0.9525 | 0.7082 | 0.1658 | 0.3450 | 0.9553 | 0.6912 | 0.2845 | 0.2845 | 0.9545 | 0.6537 | 0.3911 | 0.2358 |
| Jiaxing | 0.9554 | 0.6871 | 0.2560 | 0.3188 | 0.9541 | 0.6645 | 0.3107 | 0.3107 | 0.9537 | 0.6554 | 0.3326 | 0.3062 |
| Yiwu | 0.9336 | 0.7536 | 0.1487 | 0.2579 | 0.9291 | 0.7278 | 0.2064 | 0.2529 | 0.9246 | 0.7326 | 0.1184 | 0.3050 |
| Wuhan | 0.9607 | 0.7810 | 0.1492 | 0.2402 | 0.9597 | 0.7633 | 0.2143 | 0.2146 | 0.9467 | 0.6083 | 0.5081 | 0.1070 |
| Zhengzhou | 0.9607 | 0.7807 | 0.1352 | 0.2517 | 0.9604 | 0.7634 | 0.2148 | 0.2148 | 0.9633 | 0.7498 | 0.3361 | 0.0859 |
| Guangzhou | 0.9149 | 0.7328 | 0.0336 | 0.3413 | 0.9337 | 0.7479 | 0.2297 | 0.1952 | 0.9212 | 0.7375 | 0.0986 | 0.3053 |
| Shenzhen | 0.9092 | 0.7940 | 0.1278 | 0.1488 | 0.9082 | 0.7906 | 0.1407 | 0.1423 | 0.8921 | 0.7446 | 0.2346 | 0.1139 |
| Xiamen | 0.9350 | 0.8009 | 0.1328 | 0.1824 | 0.9283 | 0.7751 | 0.1801 | 0.1801 | 0.9253 | 0.7555 | 0.2442 | 0.1475 |
| Lhasa | 0.9984 | 0.7383 | 0.3555 | 0.1338 | 0.9985 | 0.7836 | 0.2156 | 0.2156 | 0.9982 | 0.7733 | 0.1469 | 0.2913 |
| Chongqing | 0.9921 | 0.7531 | 0.1623 | 0.3095 | 0.9924 | 0.7376 | 0.2584 | 0.2586 | 0.9919 | 0.7431 | 0.1855 | 0.3098 |
| Chengdu | 0.9605 | 0.8036 | 0.0858 | 0.2472 | 0.9632 | 0.7998 | 0.1792 | 0.1802 | 0.9613 | 0.7784 | 0.2443 | 0.1506 |
| Kunming | 0.9913 | 0.8288 | 0.1456 | 0.1868 | 0.9896 | 0.7907 | 0.2037 | 0.2041 | 0.9899 | 0.8039 | 0.1660 | 0.2143 |

The urban extent extraction results of seven representative cities using three methods were illustrated in Figure 7. These cities range from mega cities (e.g., Beijing, Shanghai, and Guangzhou), middle-size cities (e.g., Wuhan) to small cities (e.g., Lanzhou, Urumqi, and Kunming). In order to clearly present the difference between the extracted results and the reference data, the common parts, overestimated parts and underestimation parts were showed in red, green, and blue, respectively. The complete urban extent result was composed of the common parts (red pixels) and the overestimated parts (green pixels). The visual comparison revealed that the urban extent extracted from NTL data by these three methods match the reference data well. Nevertheless, all the urban extent extracted by three methods existed some overestimation. This may partly be explained by the inconsistency between NTL data and Landsat 8 images in expressing the urban and the low resolution of NTL data. In addition, there were some problems with the extracted results of LOT and INNL-SVM. The results of LOT and INNL-SVM had many holes in urban inner areas and many pixels at urban edge district were mistaken as non-urban pixels. These problems were improved in our method. The urban extent

extracted by our method had high spatial continuity and higher consistency with the reference data at urban edge district. Overall, these results indicate that our method performs well in extracting urban extent from NTL data, enhances the internal spatial continuity of the extracted results, and improves the underestimation of the edge pixels.

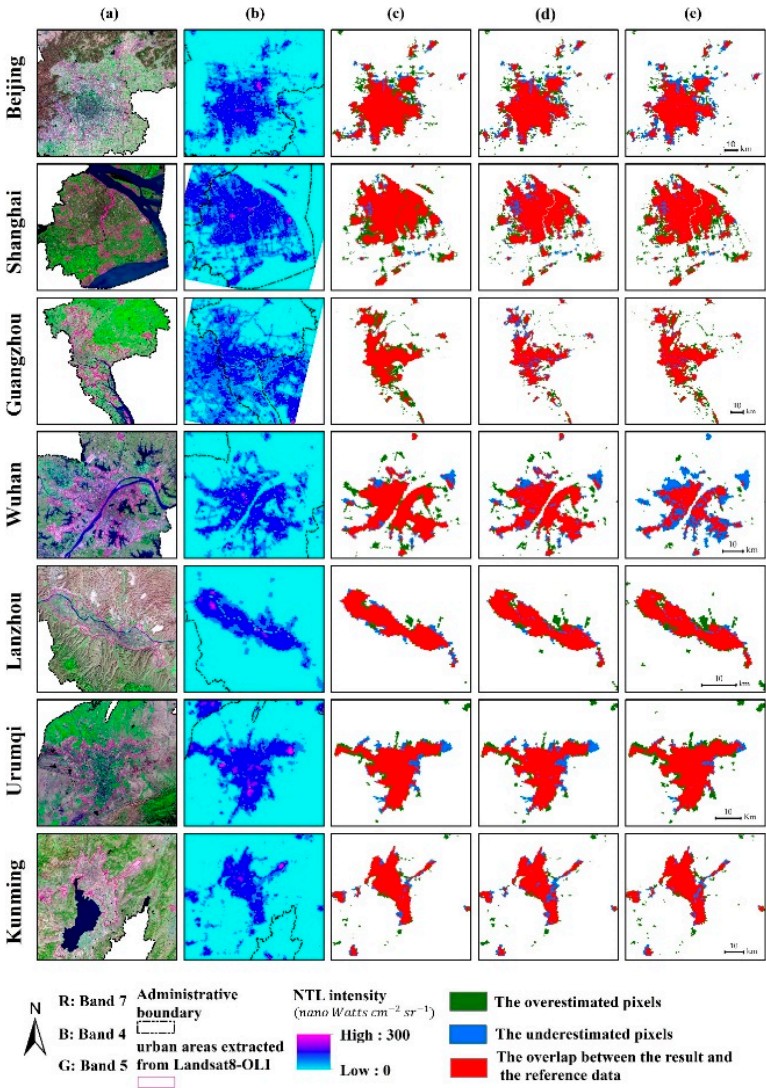

**Figure 7.** Urban extent extraction results: (**a**) Landsat8 OLI Images of selected cities, (**b**) VIIRS NTL images of selected city, (**c**) the urban extent extracted using our method, (**d**) the urban extent extracted using local optimal threshold (LOT), (**e**) the urban extent extracted using INNL-SVM.

*4.3. Comparison with Other Products*

We also compared our results in the year 2015 with the existing global urban datasets, the GAIA (2015), the MODIS500 (2015), the NUACI-based GUL (2015), the FCN (2016). Figure 8 shows the comparisons in seven representative cities, in which GAIA and NUAIC-based GUL had 30 m spatial resolution and our result, MODIS500 and FCN had 500 m spatial resolution. When visually compared with the referenced urban extent and Landsat8 OLI images, our results and FNC provided a good delineation of urban extent, while GAIA and NUACI-based GUL demonstrate the finest spatial details of urban land. The urban spatial pattern of these five dataset MODIS500 had relatively strong similarity. MODIS500 overestimated urban land for cities such as Beijing, while had more omission errors for the cities of Urumqi and Kunming. We resampled these urban datasets for the seven representative cities to a spatial resolution of 500 m on the Albers Conical Equal Area projection and calculated urban pixels and the kappa coefficient (Table 3). The Kappa value of our method and FCN ranged from 0.7 to

0.8, while that of GAIA, NUACI-based GUL, and MODIS500 spanned 0.5 to 0.7. GAIA had the highest urban area for all cities because it is an artificial impervious surface product, not only covering urban areas but also non-urban areas.

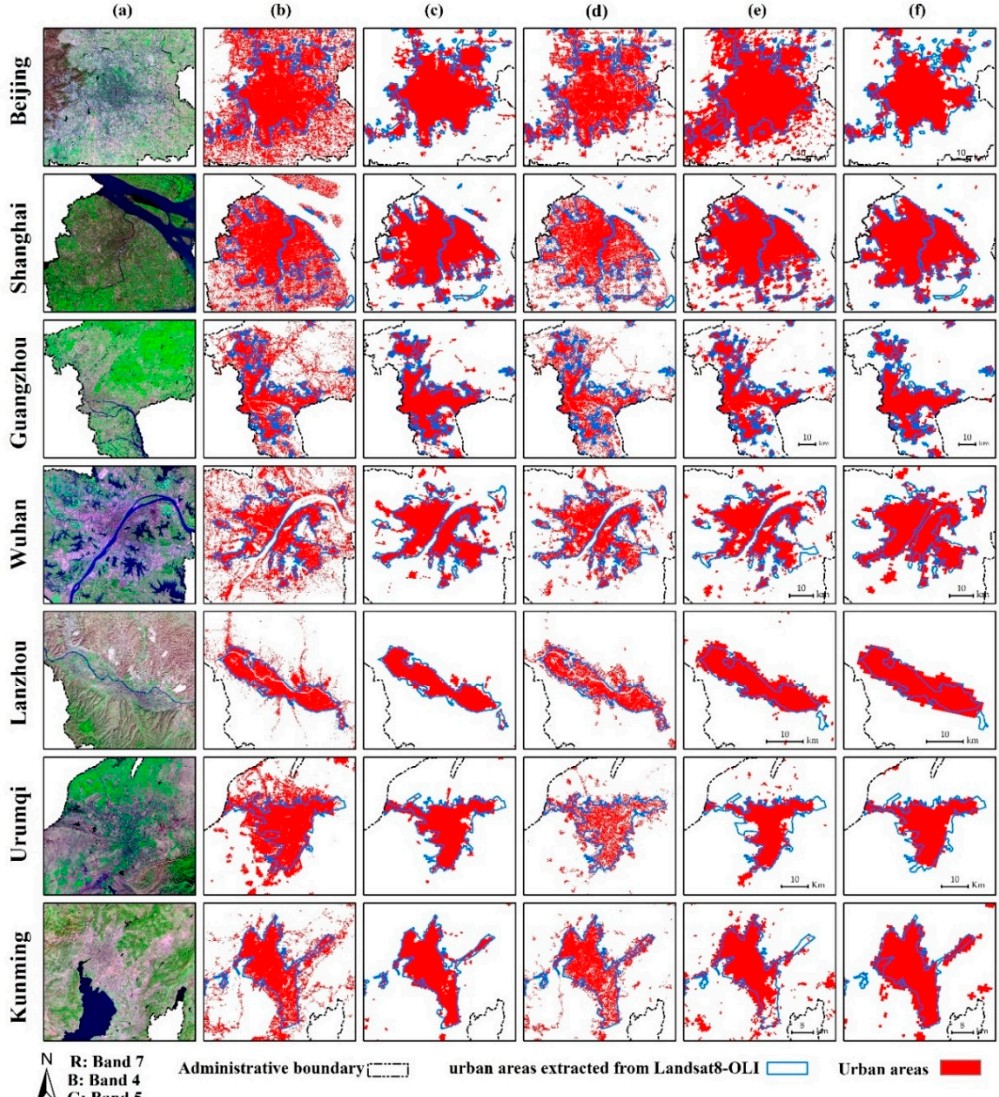

**Figure 8.** Comparing the urban pattern and spatial details between our results and other products (**a**) Landsat8 OLI Images (30 m), (**b**) the global artificial impervious area (GAIA) (30 m), (**c**) our results (500 m), (**d**) the global urban land based on a normalized urban areas composite index (NUACI-based GUL) (30 m), (**e**) the MODIS 500 (500 m), (**f**) the global urban expansion product based on a fully convolutional network (FCN) (500 m).

**Table 3.** Comparing our results in the year 2015 with other products.

| City | Our Method | | GAIA | | NUACI-Based | | MODIS500 | | FCN | | Landsat |
|------|------|-------|------|-------|------|-------|------|-------|------|-------|---------|
| | Area | Kappa | Area | Kappa | Area | Kappa | Area | Kappa | Area | Kappa | Area |
| Beijing | 8443 | 0.7978 | 12,935 | 0.5834 | 10,201 | 0.7012 | 14,085 | 0.6269 | 7712 | 0.7982 | 7736 |
| Shanghai | 10,737 | 0.7879 | 11,155 | 0.6075 | 6744 | 0.6067 | 12,332 | 0.7494 | 10,630 | 0.8119 | 9181 |
| Guangzhou | 6847 | 0.7328 | 6393 | 0.6124 | 4306 | 0.6165 | 6154 | 0.7137 | 5490 | 0.7440 | 4667 |
| Wuhan | 3610 | 0.7810 | 4163 | 0.5759 | 2883 | 0.6850 | 3719 | 0.7065 | 4255 | 0.7214 | 3224 |
| Lanzhou | 820 | 0.8256 | 1309 | 0.6376 | 1085 | 0.6725 | 1825 | 0.5308 | 1242 | 0.7083 | 887 |
| Urumqi | 1606 | 0.8263 | 2963 | 0.6003 | 849 | 0.4396 | 1521 | 0.7722 | 1578 | 0.8137 | 1575 |
| Kunming | 2259 | 0.8288 | 3016 | 0.6524 | 1920 | 0.6168 | 3685 | 0.5677 | 3519 | 0.6967 | 2150 |

Note: "Area" means the number of urban pixels in spatial resolution 500 m, and "Landsat" means the urban extent obtained from Landsat.

## 5. Discussion

### 5.1. The Proposed Method Can Effectively Extract Urban Extent from NPP-VIIRS NTL

This study proposed an automatic and reliable approach to extract urban extent from NTL data quickly and accurately. Different neighborhood sizes of spatial context constraints are utilized to characterize the edge and interior of the urban areas and reduce the effects of noise and abnormal light intensity. The spatial context constrains can enhance the signals of urban areas and weaken the influence of non-urban areas light. The major difference from previous studies is that we attempt to develop a simple method without the help of ancillary data and transform urban extent extraction as an urban and non-urban pixels-based clustering problem, which could avoid the tedious procedure of threshold or sample selection. The experimental results prove that the proposed method is reliable and robust, which can extract the urban extent of different areas with different regional environment and social and economic development level accurately. The kappa ranged from 0.687 to 0.829 with an average of 0.768 (1.80% higher than LOT and 4.88% higher than INNL-SVM). The extraction results of the proposed method demonstrate great integrity and connectivity, which are consistent with the reference data. Comparison with LOT and INNL-SVM showed that the proposed method could guarantee high extraction accuracy, while increasing detail information of areas in edge district (Figure 9a) and reducing errors affected by low brightness of areas in urban inner (Figure 9b). Among three popular methods, LOT requires ancillary data to determine optimal thresholds and the extraction results of INNL-SVM are highly rely on sample data. However, for large-scale studies, considerable time and labor are needed to obtain high-accuracy ancillary data and sample data, which limits the practicability of LOT and INNL-SVM. The proposed method automatically recognizes urban extent depending on NTL data itself, without referring to other data, increasing its efficiency for extracting urban extent at large scales or long-term temporal scales. These prove that the proposed method has extensive applicability in the field of urban extent extraction.

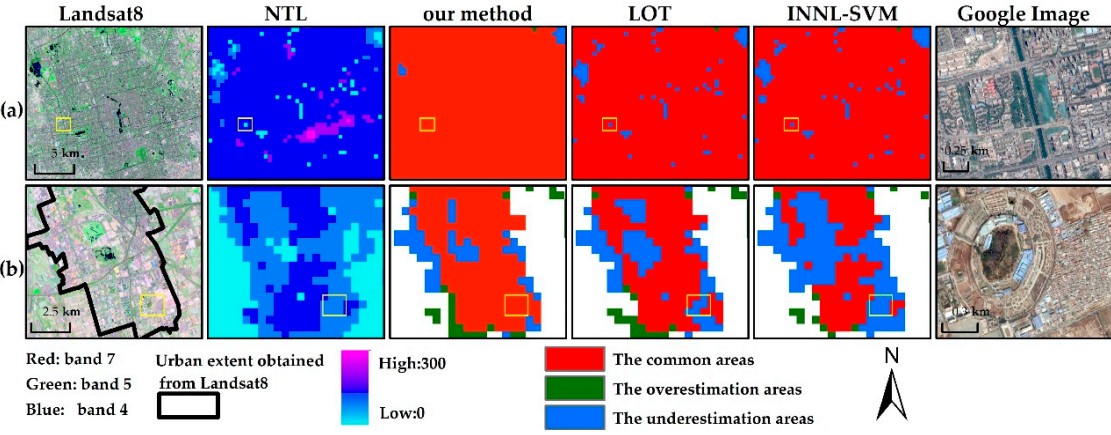

**Figure 9.** Spatial Comparison of the results different methods for Beijing: (**a**) the inner part of urban; (**b**) the edge district of urban; the yellow box is the area zooming in of the google image.

In addition, we applied our method to extracting urban extent in the years 2013 and 2018 for the seven representative cities to demonstrate the potential of our method in urban dynamic analysis. We selected the GAIA to comparison because of its appropriate time coverage and relatively high resolution. Figures 10 and 11 showed urban expansion of Shanghai and Lanzhou from 2013 to 2018. The urban extent extracted using our method in the years 2013, 2015, and 2018 revealed the urban growth trace and the growth regions of our method matched to those of GAIA to certain degrees considering the difference between imperviousness- and NTL- indicated urbanization.

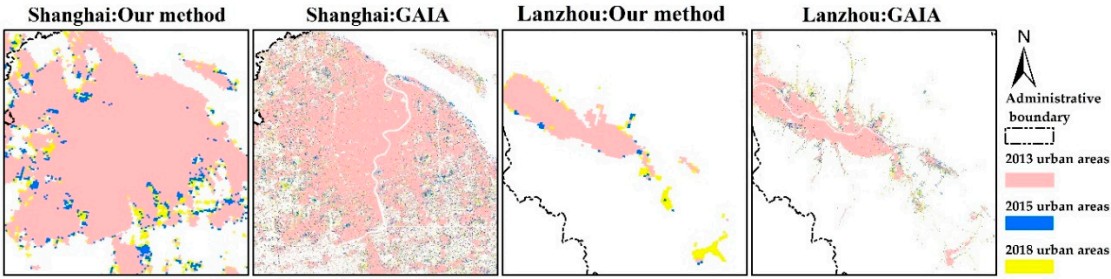

**Figure 10.** Urban expansion from 2013 to 2018 for Shanghai and Lanzhou.

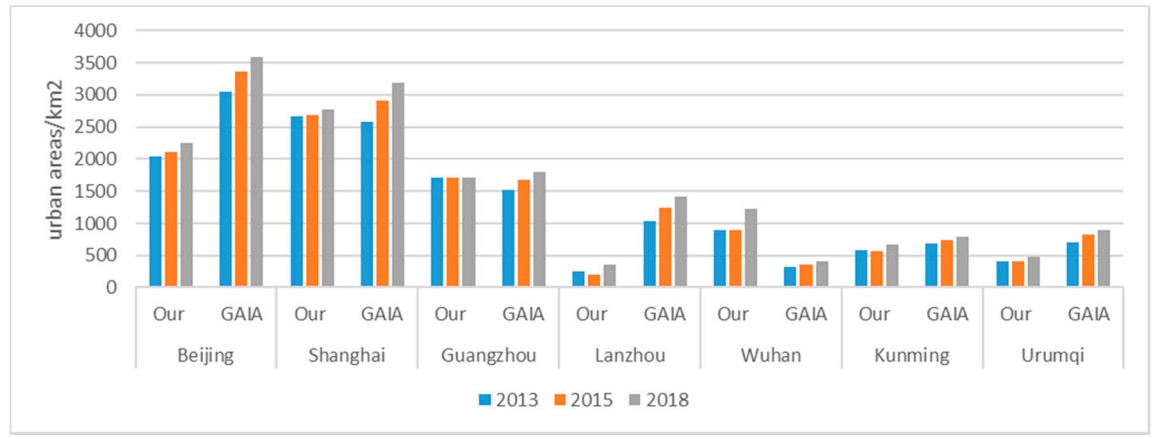

**Figure 11.** Growth of urban areas from 2013 to 2018.

## 5.2. The Disadvantages of the Proposed Method

There were some overestimation and underestimation of our extraction results. For example, Lhasa and Hohhot had relatively high OE and Ningbo and Guangzhou had relatively high CE. We compared Lhasa and Urumqi because they both were in west China, while Urumqi had much lower OE and CE. The brightness distribution in Urumqi was relatively uniform and urban areas were noticeably brighter than non-urban areas in urban edge district, while the brightness distribution in Lhasa showed obviously spatial aggregation (Figure 12). The great brightness difference between urban center and the surrounding areas resulted in mistaking many urban areas for non-urban areas. Observation of incorrect urban areas illustrated that the inconsistent description of urban areas between NTL data and Landsat images was one of the main causes of overestimating the urban areas. In Landsat, area A in Figure 13 was regarded as a non-urban area because of its low percentages of impervious surfaces, while NTL identified it as an urban area because of its relatively high brightness. From the google image, there are several roads through area A and some high-density building areas are embedded in contiguous farmland (Figure 13). It seems that there is a close relationship between area A and the main urban areas.

In general, the proposed method has the following shortcomings. It is based solely on NTL data result in some misclassification. The overflow effect of NTL data and the strong light from non-human active area facilities (such as city periphery, airport, oil field) may cause the overestimation of urban areas. The overpass time of VIIRS is near 1:30 a.m., which leads to underestimation in places with declining brightness in midnight. In addition, the proposed method uses a simple way to represent the spatial context information and classify urban pixels, which is not sufficient for complex areas. In the area of rapid urban development and continuous emergence of small towns, it is more likely to exist overestimation between cities and towns. In the area where human activities are mainly concentrated in the center part, it is easy to be underestimated in the edge district. Besides, this study remains narrow in focus dealing only with the NPP-VIIRS data. The developed method is not suitable for other NTL products due to the different of spatial resolution, radiation resolution and process

procedures. To extract urban extent from other NTL products such as DMSP-OLS and Luojia-01, the appropriate clustering method based on spatial context constraints should be designed according to the characteristics of the NTL product. Future research might focus on developing the unsupervised urban extent extraction model with better spatial features representation capabilities to adapt to different NTL products and improve the accuracy of complex environmental areas. Consider using convolution neural network methods to enhance the ability of spatial information expression and integrating other data sources (such as the human activity data) to descript urban areas more comprehensively.

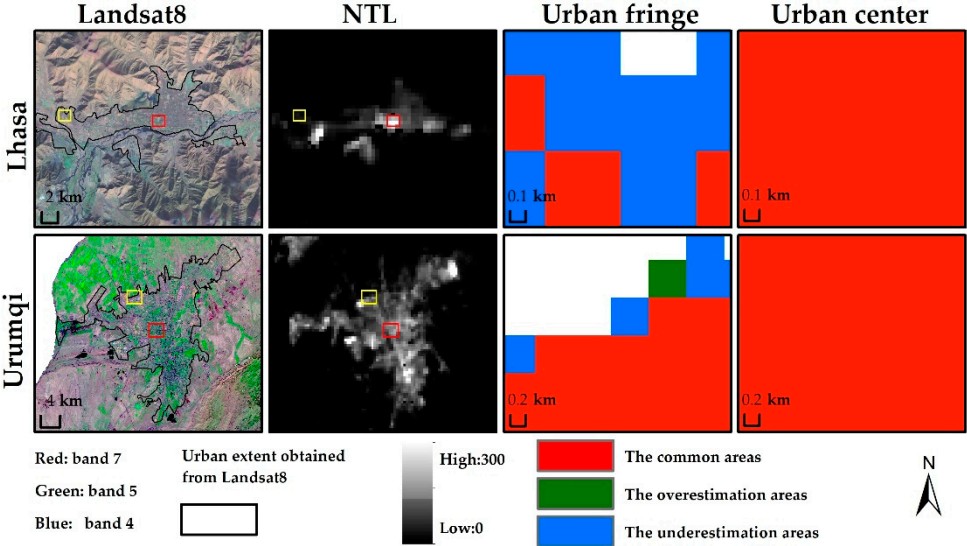

**Figure 12.** The spatial distribution of nighttime light intensity in Lhasa and Urumqi (yellow and red boxes indicate areas zoomed in column 3 and 4, respectively).

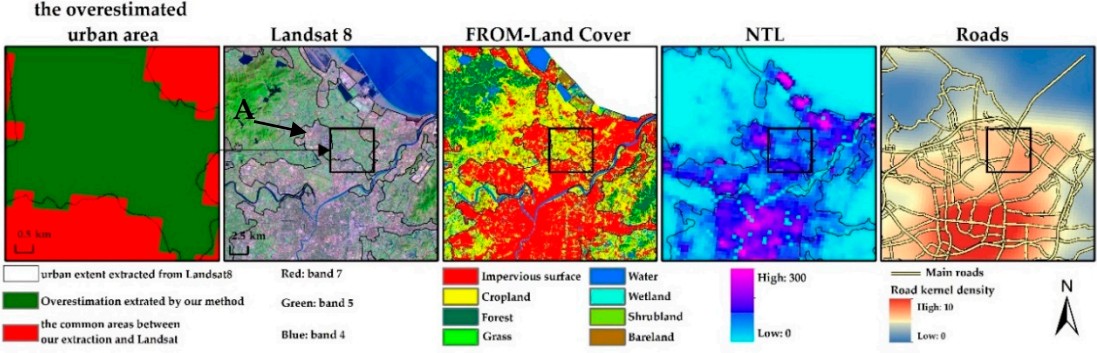

**Figure 13.** Analysis of overestimation urban area for Ningbo: (black box is the overestimation area zoomed in column 1, which was indicated by "A" in text).

*5.3. The Sensitivity Analysis of Parameters*

The algorithm of the proposed method was presented in Supplementary Table S1, where there are three free parameters: R1, R2, and N. The parameters R1 and R2 are used to determined the filtering window radius for step 1 and step 2 (Figure 2), respectively. The window size of step 1 is $(2 \times R1 + 1) \times (2 \times R1 + 1)$, and the window size of step 2 is $(2 \times R2 + 1) \times (2 \times R2 + 1)$. The parameter N is used to specify the threshold value for urban edge district detection.

A sensitivity analysis was performed for R1, R2, and N to examine how changes in these parameters impact urban extent extraction accuracy. We use the average kappa coefficient of 25 experiment cities to represent the extraction accuracy. Figure 14 showed that the accuracy kept in the range of 0.68 to 0.8 with fewer outliers, which indicated that no matter how the parameters change, our results keep a relative high accuracy. However, parameters also can have important impacts on the accuracy.

The accuracy increased with the increase of N, and it achieved the highest value when N was 1. For R2, the accuracy had the highest value at 4 or 5. For R1, the results had the highest value at 2. Besides, we found that these parameters have little influence on each other.

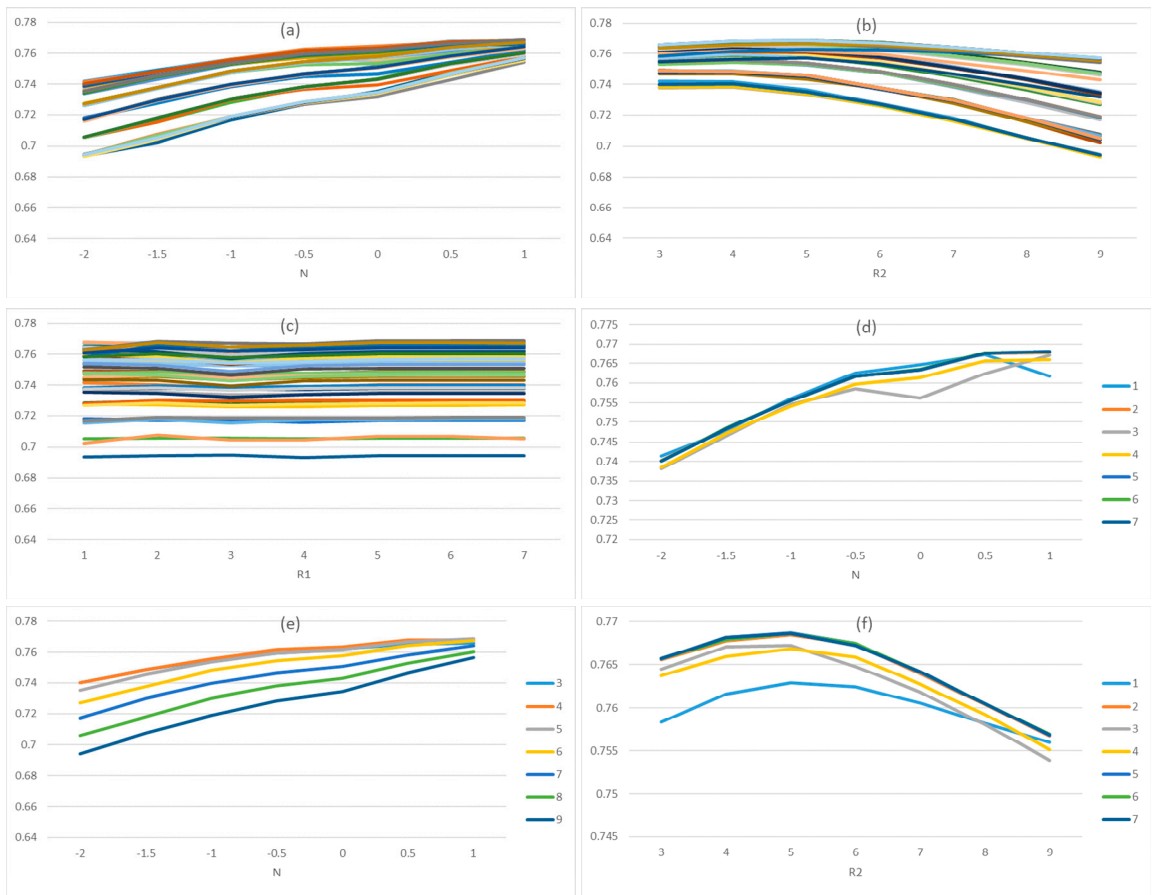

**Figure 14.** The Sensitivity analysis of parameters: (**a**–**c**) were the results of all parameter combinations; (**d**) was the results withR2 = 4; (**e**) was the results with R1 = 2; (**f**) was the results with N = 1.

## 6. Conclusions

This study set out to develop an unsupervised method for urban extent extraction from NTL data, avoiding the time-consume preparation of ancillary or labelled data. The proposed method applies the pixel-based clustering algorithm to distinguish urban areas from background. Multi-scale spatial context constraints are utilized to integrate surrounding information of pixels, to increase the distinction between urban pixels and nonurban pixels and correct those areas with abnormal light intensity. The experiments confirmed that it has high accuracy and can extract urban extent with different regional environments and socioeconomic development levels (the kappa of the proposed method ranged from 0.687 to 0.829, while the kappa range of LOT and INNL-SVM was 0.664 to 0.811 and 0.608 to 0.821, respectively). The overlay between extraction results and reference data demonstrates that the urban extent extracted by the proposed method had greater spatial integrity and connectivity in urban inner and lower omission errors in urban edge district. The method can delineate urban extent accurately and automatically, without the time-consuming and laborious process of threshold selection and training dataset construction, which saves a large amount of time and labor. These indicate the broad application prospects of the proposed method in large-scale urban research. In addition, the results in this study also proved the feasibility of applying unsupervised methods for urban extent extraction, which provides new inspiration for urban extent extraction based on NTL data and has a number of important implications for future practice.

**Supplementary Materials:** The following are available online at http://www.mdpi.com/2072-4292/12/22/3810/s1.

**Author Contributions:** Conceptualization, F.Z.; Funding acquisition, F.Z., Z.D. and R.L.; Methodology, X.C.; Project administration, R.L.; Writing—original draft, X.C.; Writing—review & editing, F.Z. All authors have read and agreed to the published version of the manuscript.

**Funding:** This research was funded by the National Key R&D Program of China (2018YFB0505000, 2017YFB0503604), National Natural Science Foundation of China (41671391, 41922043, 41871287).

**Conflicts of Interest:** The authors declare no conflict of interest.

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
