# Peer review of "An Unsupervised Urban Extent Extraction Method from NPP-VIIRS Nighttime Light Data"

_remotesensing, doi:10.3390/rs12223810_

Round 1
Reviewer 1 Report
The authors attempt to define the urban extent from nighttime light - which is a popular topic among all nighttime light related researched - with a combination os machine learning techniques, including K-means clustering and classification. The result shows that the method proposed in this paper have less over/under-estimated pixels compared to methods proposed in previous studies, i.e. LOT and INNL-SVM.
The manuscript is well laid out, nevertheless some sections are not described clear enough, thus it causes difficulties for readers to comprehend the result and conclusion. Hence it requires a major revision before one can fully review and comprehend the full content of the manuscript.
Here are some comments for the authors:
[1] Table 1: The authors aims to use numbers showing population and GDP for the 25 sample cities to display their variety. The population and GDP is however, not being used in the method and other places in the manuscript. In addition, the numbers are very off from the actual figure. The population is 100 times larger than the actual number, while GDP is 10 times larger. I believe the mistake is caused by the different unit used in the static report and in the table.
[2] All maps has the strange +/- mark. I am unclear what that means.
[3] Line 134: Earth Observation Group (EOG) which makes NTL data is now part of the Payne Institute for Public Policy in Colorado School of Mines.
[4] Line 136: ..."which has been multiplied by 1E9" is unnecessary here.
[5] Section 3.2: This section is very critical in the manuscript and deserves more clarification.
First of all, "Silhouette coefficient" is not clearly described, and it is unclear what the "object" is used to match with the cluster.
Second, the authors had already decided to have K=2 as described in section 3.1, and use them as "urban" and "non-urban". Then why perform multi-K analysis as shown in Figure 3? Also it is better to show the clustering result as a map for readers to better comprehend how the clustering works to recognize urban and non-urban regions.
Third, what is the "law of light intensity distribution"? Being named as a law must be true globally, as a not so well-known law, this deserved far more description.
Fourth, Eq (5), the "<" mark for Pnon-urban inner should be ">"? How is the Pedge areas different in Purb and Pnon?
[6] Line 262: "...And Dou..." should be "...and Dou..."
[7] L268: What are PA and UA used here? They are not explained before hand.
[8] L291: What is the difference between "urban edge district" and "urban edge pixels"?
[8] Figure 5: The texts are too small.
[9] Figure 8: The Landsat 8 column cells are inconsistent. The area with black outlined polygon should be transparent filled, while the upper cell does not have the urban area marked.
Reviewer 2 Report
I think this is an excellent manuscript deserving of swift publication.
Reviewer 3 Report
This study developed an unsupervised method to extract the urban extent from VIIRS nighttime light data using the k-means clustering algorithm. The proposed method was validated using the extents of 25 Chinese cities derived from Landsat 8 images, and further compared with two common methods of LOT and INNL-SVM. However, there are several major concerns about the effectiveness and robustness of the proposed method.
-Major points
- My first concern is the way of evaluation. The accuracy of the extracted urban extent was assessed using the “ground truth” data (i.e. urban extent obtained from Landsat 8 images). However, the authors did not specify the extraction methods or correctness of the “ground truth” data, which could have a certain impact on the evaluation of results. Since NTL and Landsat are very different images, I doubt the reliability of comparing urban extents derived from these two images directly. One possible evaluation approach is to randomly identify some urban and non-urban pixels in the study area and then verify the classification results using those validation samples. Furthermore, given there are many global or national urban/impervious surface products released recently, I suggest using additional datasets as a comparison or replacement. Below are several related studies for reference:
Gong, Peng, Xuecao Li, and Wei Zhang. "40-Year (1978–2017) human settlement changes in China reflected by impervious surfaces from satellite remote sensing." Science Bulletin 64.11 (2019): 756-763.
Gong, Peng, et al. "Annual maps of global artificial impervious area (GAIA) between 1985 and 2018." Remote Sensing of Environment 236 (2020): 111510.
Liu, Xiaoping, et al. "High-spatiotemporal-resolution mapping of global urban change from 1985 to 2015." Nature Sustainability (2020): 1-7.
- My second concern is the performance of the proposed method. Two commonly used methods, LOT and INNL-SVM, were adopted for a comparative purpose. However, the proposed method was not much better than these two methods in terms of OA and CE (Table 2). This means that the proposed method had an overestimation of urban areas. Spatial distributions of the overlap and mismatch areas between the derived result and the reference data also showed some overestimated and underestimated regions, which were mostly distributed in urban fringes (Figure 6). As for the Kappa coefficient and OE, the results were better than those of LOT and INNL-SVM, but not significantly enhanced. Overall, there is still room for improvement in urban edge pixels classification.
- Lastly, I wonder if this method can be applied to track time-series urban dynamics. NTL data has a long archive of nearly three decades (from 1992 to present). Recent advances have shown the potential of a harmonized NTL dataset from the calibrated DMSP and VIIRS data in mapping annual urban dynamics. I recommend the authors to address this issue in the Discussion.
Minor issues:
Line 106, Does “NTL data” here refer to “NPP-VIIRS NTL data”? If it does, I suggest clarifying this term because the methods developed for VIIRS data are different from those for other NTL data such as DMSP. In the introduction part, it is necessary to describe clearly for a better understanding of broader readers.
Line 113, Check the grammar issue.
Line 122-124, Where do statistics of population and GDP come from, and how is these statistics used for? Please add references.
Line 151, Please specify the algorithm for atmospheric correction of Landsat 8.
Line 154, What is the source of “high-resolution images” (for example, Google Earth)? The description is too abstract.
Line 199-200, Both high-intensity and low-intensity groups refer to non-urban pixels. Please address the issue.
Line 209, Should be “where E is the loss function...”. Same to the equations below.
Line 214-215, Again, please provide the reference for Silhouette coefficient.
Line 221-222, As you pointed out, due to the low light intensity of water and green spaces within urban areas, light values observed in these areas are relatively low and reasonable. I doubt whether it is appropriate to use expressions such as "correct abnormal light values". Besides, what is your definition of the urban area?
Line 222-224, Maybe you could do a comparative analysis to show the effectiveness of the median filter compared with other types such as Gaussian filter or mean filter.
Line 236, “Pnon” should be “Pnon”. Moreover, in Eq. 5, the definition of ????? ????? should be where ????? ????? > ????−???_???? + ????−???_???.
Line 262, “And” should be “and”.
Line 347-353, I think a discussion on the influence of input image sizes is unnecessary here. The focus of this article should be the validity, reliability, and robustness of the proposed method for extracting the urban extent. So I suggest removing section 5.1.
Reviewer 4 Report
The authors present a method for urban extent extraction in remote sensing imagery. The content of the paper is interesting, however, i have several concerns:
- The methodology is not novel as it relies on simple algorithms. The study is purely experimental.
- I think the authors should investigate other recent methods to increase the contribution level such as advanced clustering algorithms and so.
- It not clear the connection between the different parts of the method.
- Please provide an algorithm that details the method with its free parameters.
- A sensitivity analysis should be presented in addition to comparisons to state-of-the-art methods.
- What about the results if auxiliary data are used.
Round 2
Reviewer 1 Report
The manuscript is significantly improved after revise. Suggest publish as is.
Reviewer 4 Report
I do not have additional comments.